# Complete transition from chromosomal to cytoplasmic sex determination during prolonged *Wolbachia* symbiosis

Takahiro Fukui[1,2], Tomohiro Muro [1], Noriko Matsuda-Imai [1], Tatsunori Kaneda[1], Hidetaka Kosako [3], Hideaki Hiraki[4], Keisuke Shoji [5], Takeshi Fujii [6], Yutaka Suzuki [7], Atsushi Toyoda [8], Takehiko Itoh [4], Takashi Kiuchi [1] & Susumu Katsuma [1] ✉

*Wolbachia* infection causes male-specific death in *Ostrinia furnacalis*, but its removal from infected strains results in female-specific death instead of restoring 1:1 sex ratio, suggesting that cytoplasmic *Wolbachia*, not the host genome, primarily determines femaleness in infected strains. This phenomenon is a striking example of the evolutionary outcome of cytoplasmic sex determination, potentially arising from prolonged host-symbiont co-evolution. Although we recently identified Oscar, the *Wolbachia*-encoded male-killing effector targeting the host masculinizing factor OfMasc in *Ostrinia* moths, inactivation or loss of the host's endogenous feminizer remains unknown. Here we identify a W-linked primary feminizer, *OfFem* piRNA, which targets an mRNA encoding an OfMasc-interacting protein Ofznf-2. We demonstrate that Ofznf-2 is essential for both masculinization and dosage compensation. We also show that *OfFem* piRNA is entirely absent in the *Wolbachia*-infected lineage, providing molecular evidence that a male-killing *Wolbachia* hijacks the host feminizing piRNA function by acquiring the Oscar protein during prolonged endosymbiosis.

Some bacterial endosymbionts, such as *Wolbachia* species, manipulate the reproductive systems of arthropod hosts through male killing, feminization, cytoplasmic incompatibility, or parthenogenesis[1]. In various host-endosymbiont combinations, male killing causes infected mothers to produce only female progeny by selectively killing the male offspring[2]. In the interaction between *Ostrinia* species and male-killing *Wolbachia* strains (wSca or wFur, Fig. S1), an evolutionary phenomenon termed "cytoplasmic sex determination" has been observed, as in a few other arthropod species[3–5]. In wFur-infected *Ostrinia furnacalis*, the

*Wolbachia* male-killing protein Oscar essentially determines femaleness, as the host's endogenous feminizing factor is considered to have been inactivated or lost during prolonged symbiosis[6–9]. While several bacterial effectors responsible for reproductive manipulations have recently been identified[9–12], no molecular evidence has yet demonstrated the inactivation or absence of the host's endogenous feminizing factor in symbiont-dependent cytoplasmic sex determination systems.

Lepidopteran sex chromosomes are female-heterogametic, with female-specific W chromosomes that are enriched in repetitive

[1]Department of Agricultural and Environmental Biology, Graduate School of Agricultural and Life Sciences, The University of Tokyo, Bunkyo-ku, Tokyo, Japan. [2]Graduate School of Science, Chiba University, Chiba, Chiba, Japan. [3]Division of Cell Signaling, Fujii Memorial Institute of Medical Sciences, Institute of Advanced Medical Sciences, Tokushima University, Tokushima, Japan. [4]School of Life Science and Technology, Institute of Science Tokyo, Meguro-ku, Tokyo, Japan. [5]Graduate School of Bio-Applications and Systems Engineering, Tokyo University of Agriculture and Technology, Koganei-shi, Tokyo, Japan. [6]Faculty of Agriculture, Setsunan University, Hirakata, Osaka, Japan. [7]Department of Computational Biology, Graduate School of Frontier Sciences, The University of Tokyo, Kashiwa, Chiba, Japan. [8]Comparative Genomics Laboratory, National Institute of Genetics, Mishima, Shizuoka, Japan. ✉e-mail: skatsuma@g.ecc.u-tokyo.ac.jp

elements and transposons[13–16]. These elements serve as sources of PIWI-interacting RNAs (piRNAs), a class of small RNAs that silence transposons[17]. In *Bombyx mori* and *Plutella xylostella*, female-specific piRNAs originating from the W chromosome, called *Feminizer* (*Fem*) piRNAs, determine femaleness[18,19]. *Fem* piRNAs target an mRNA of *Masculinizer* (*Masc*), a lepidopteran-specific CCCH-tandem zinc finger protein gene essential for both masculinization and dosage compensation[18,20–30]. The *Fem* piRNA-PIWI protein complex cleaves *Masc* mRNA[18], inhibiting male-type splicing of *doublesex* (*dsx*) and dosage compensation in male embryos[18,31], making *Fem* piRNAs the primary determinant of sex in Lepidoptera. In contrast, we discovered that in *O. furnacalis*, *Masc* (*OfMasc*) is not regulated by female-specific piRNAs during sex determination, indicating a unique sex determination system distinct from that of *B. mori*[32].

In Japan, the infection rate of *Wolbachia* in *O. furnacalis* is relatively low[33] (~15%), where infected individuals consistently harbor a single strain of *w*Fur (*w*Fur9)[33,34]. Because this infection causes complete male killing, where *Wolbachia*-infected females can only mate with uninfected males, the autosomal gene pool is shared between uninfected and *Wolbachia*-infected strains. Unlike male killing observed in other lepidopteran insects[35–39], depleting *w*Fur from infected moths using antibiotics induces female killing instead of restoring a 1:1 sex ratio[6,8]. This suggests that the primary feminizing factor, which may be located on the matrilineally inherited W chromosome, has been lost or inactivated in the *Wolbachia*-infected strain.

Recently, we demonstrated that the *Wolbachia* male-killing factor Oscar degrades OfMasc, thereby inhibiting male-type splicing of *O. furnacalis dsx* (*Ofdsx*) and dosage compensation[9,40]. Ectopic overexpression of *Masc* in females has been shown to cause female-specific death during larval stages[41,42], as observed in female killing in antibiotics-treated *Wolbachia*-infected *O. furnacalis*, suggesting that Oscar directs the female-determining pathway by inactivating OfMasc in infected strains where the primary feminizing factor may be missing or nonfunctional. To unravel the molecular mechanism underlying *Wolbachia*-dependent cytoplasmic sex determination, identifying the endogenous feminizing factor of *O. furnacalis* and validating its absence or inactivation in infected females is essential.

Here, we identify a single feminizing piRNA derived from the W chromosome of *O. furnacalis*. Unlike *Fem* piRNAs in *B. mori* or *P. xylostella*, this piRNA targets an mRNA encoding Ofznf-2, a protein that interacts with OfMasc. Knockdown experiments reveal that *Ofznf-2* is crucial for male-type *dsx* splicing and dosage compensation. Additionally, we find that this piRNA is entirely absent in the *Wolbachia*-infected lineage of *O. furnacalis*. These findings provide molecular evidence that a male-killing *Wolbachia* hijacks the role of a W-linked feminizing piRNA by acquiring the Oscar protein during prolonged endosymbiosis.

## Results

### A single W-linked piRNA targets mRNAs for CCCH-type zinc finger proteins in *O. furnacalis*

We hypothesized that the feminizing factor in *O. furnacalis* had been inactivated or lost in the *Wolbachia*-infected lineage. To identify the primary feminizing factor in *O. furnacalis*, we sequenced the transcriptomes of molecularly sexed early embryos during the period of sex determination[32]; the sex-specific splicing of *Ofdsx* is first observed after 12 hours post-oviposition (hpo) and is established at 48 hpo. From the de novo assembled transcriptome, 14 contigs were found to be significantly more highly expressed in females than in males across all embryonic stages (Fig. S2a). Among these, two contigs met the following criteria: (i) higher expression in uninfected embryos and ovaries compared to infected ones, and (ii) higher expression in uninfected ovaries compared to uninfected testes (Fig. S2b). In two in-house assembled whole-genome assemblies (version #1 and version #2, derived from *O. furnacalis* collected in Japan; Table S1) and the

published genome of *O. furnacalis*[43] (collected in China), these two contigs were mapped to a single locus on the W chromosomes. Fragments from this locus were PCR-amplified exclusively from female genome samples (Fig. S3a, b). Combined with the observation that its expression was restricted to females, not males, during the sex determination period (Figs. 1a, b, S3c, and S4a, b), this locus emerged as a strong candidate for encoding the feminizing factor. We designated this locus as *O. furnacalis Feminizer* (*OfFem*). The *OfFem* locus consisted of numerous repeat elements (Fig. S5a), a typical feature of piRNA clusters[44–46]. Small RNA sequencing (small RNA-seq) revealed that a single female-specific piRNA (*OfFem* piRNA) was abundantly produced from the *OfFem* locus in embryos and gonads (Figs. 1a, c and S4a, c).

The most plausible target gene of *OfFem* piRNA, named *Ofznf-2*, is a putative homolog of the *O. scapulalis* zinc finger gene *Oszznf-2*[47,48]. Within the *OfFem* locus, sequences of 162–197 base pairs (bp) homologous to *Ofznf-2* were repeated 17 times (Fig. S5a, b). Six loci containing *Ofznf-2* genes were identified across the *O. furnacalis* genome (version #2) (Fig. S6). Two copies of *Ofznf-2* on the W chromosome appeared pseudogenized due to a 6-kbp-long insertion (Fig. S6), so they were excluded from further analysis. The remaining four copies were categorized into two expression types: late type (*Ofznf-2-L*) and early type (*Ofznf-2-E*) (Fig. 1d, e). Additionally, *Ofznf-2-E* contained a specific 75–131 bp-long insertion absent in *Ofznf-2-L* (Fig. S6). A copy of *Ofznf-2-L* on the 28th chromosome was expressed throughout embryogenesis after 12 hpo, while two copies on the 28th chromosome and one on the Z chromosome, classified as *Ofznf-2-E*, were expressed exclusively around 12 hpo, which is the onset of sex determination[32] (Figs. 1d and S7a). Both the version #1 genome (derived from a different individual from that used for version #2 genome) and the published genome[43] also encoded at least one copy of *Ofznf-2-L* and *Ofznf-2-E*, respectively (Fig. S8, Table S2), suggesting variability in the copy number of *Ofznf-2-E* among *O. furnacalis* populations.

Across the genome, *OfFem* piRNA perfectly matched the antisense orientation of *Ofznf-2* sequences (Figs. 1e and S6). All copies of *Ofznf-2-E* and *Ofznf-2-L* encoded short proteins (126–187 aa and 107 aa, respectively) with two CCCH-type zinc finger domains (Figs. 1e and S9a). These proteins showed high homology to Bmznf-2, which is required for proper male differentiation in *B. mori*[49,50]. In a cell line from *O. scapulalis*, male-type *O. scapulalis dsx* (*Osdsx*) was expressed, but transinfection with male-killing *Wolbachia* (*w*Sca) induced the expression of female-type *Osdsx* and altered *Oszznf-2* expression[47,48]. These observations support the role of Ofznf-2 proteins in sex determination. A BLASTp search identified homologs of *Ofznf-2* across Lepidoptera, all containing two CCCH-type zinc finger domains similar to *Masc* homologs (Fig. S10). A rapid amplification of cDNA ends (RACE) experiment confirmed that *OfFem* piRNA-mediated cleavage of *Ofznf-2* mRNA occurred in early embryos (Figs. 1f and S11). At this stage, *Ofznf-2* mRNA expression was higher in males than in females (Fig. S7b, c). These findings suggest that *OfFem* generates a female-specific piRNA that suppresses *Ofznf-2* mRNA, which may play a critical role in male-determining functions.

### *Ofznf-2* is required for male determination in *O. furnacalis*

To test whether *Ofznf-2* is essential for male determination, we performed embryonic knockdown experiments using small interfering RNA (siRNA) designed to target the conserved sequence shared across all copies of *Ofznf-2*. Microinjection of *Ofznf-2* siRNA inhibited the male-type splicing of *Ofdsx* (Fig. 2a, b) and led to the upregulation of Z-linked genes in egg masses (Fig. 2c). These findings indicate that *Ofznf-2* is necessary for both male-type splicing of *Ofdsx* and dosage compensation. Furthermore, embryonic knockdown of *Ofznf-2* produced adult moths with a female-biased sex ratio (Fig. 2d), suggesting that *Ofznf-2* plays a critical role in male viability. All observed phenotypes mirrored those resulting from the knockdown or degradation of OfMasc[8,9,21,32,40]. Additionally, two of the 26 male adults that emerged from eggs injected

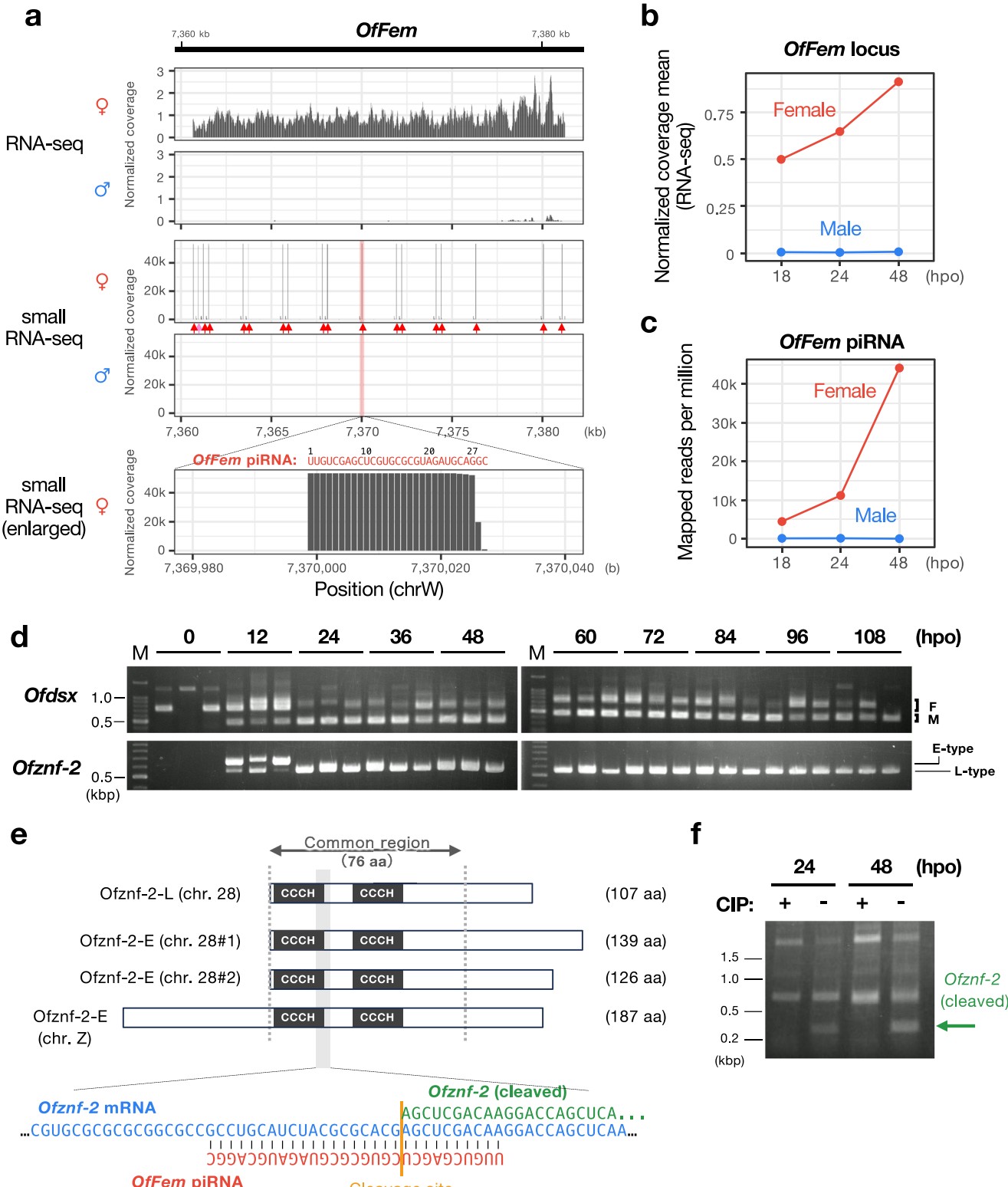

**Fig. 1 | A single female-specific *OfFem* piRNA targets *Ofznf-2* mRNA in *Ostrinia furnacalis*. a** Mapping of RNA-seq and small RNA-seq reads on the *OfFem* locus in sexed eggs (48 hpo). Arrows indicate the positions of the *OfFem* piRNA-producing sites. Red arrows denote exact matches to *OfFem* piRNA, and a pink arrow shows a sequence with a single nucleotide mismatch. **b** Expression profile of *OfFem* transcripts in sexed eggs during the sex determination period. **c** Expression profile of *OfFem* piRNA in sexed eggs during the sex determination period. **d** Splicing patterns of *Ofdsx* mRNA and expression of *Ofznf-2* variants during embryogenesis (*n* = 3). **e** Structures of Ofznf-2 variants annotated in the version #2 genome (top), with the putative *OfFem* piRNA-mediated cleavage site indicated by a yellow line (bottom). **f** Cleavage site identification of *Ofznf-2* mRNA through RACE analysis of mRNA fragments without calf intestinal phosphatase (CIP) treatment. Nucleotide sequences of the cloned fragments are detailed in Fig. S11. Similar results were obtained in two technical replicates. Source data are provided as a Source data file.

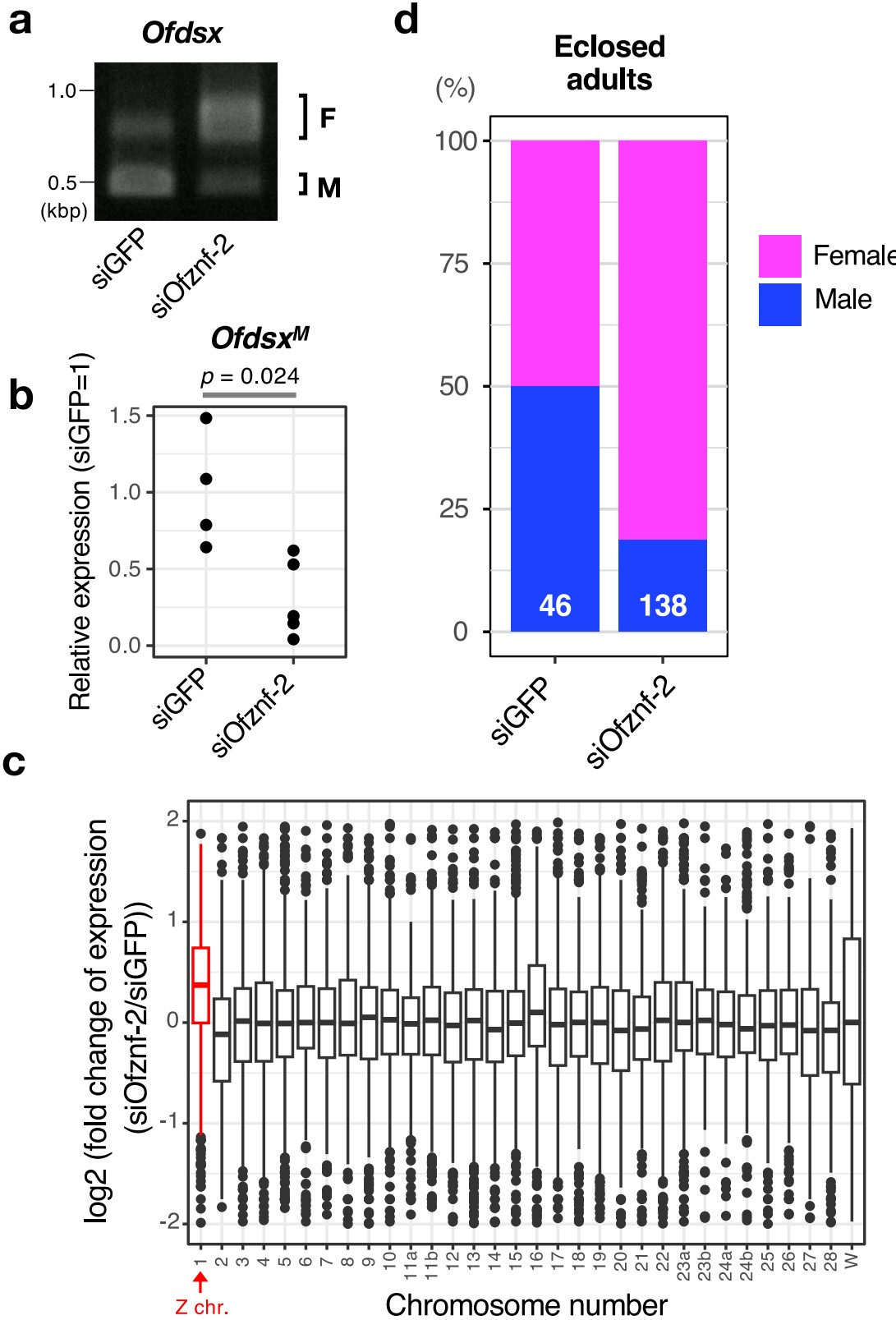

with *Ofznf-2* siRNA displayed female-like wing morphology (Fig. S12), further supporting the role *of Ofznf-2* in masculinization.

### Ofznf-2 cooperates with OfMasc in the masculinizing pathway in *O. furnacalis*

Given that *OfMasc* and *Ofznf-2* are both required for male determination in *O. furnacalis*, we hypothesized that these proteins physically interact. To test this, we used a cell line derived from *O. furnacalis* embryos[9]. Expression of GFP-tagged OfMasc or OfMascNLS (a derivative of OfMasc lacking the active nuclear localization signal but retaining masculinizing activity[51]) in the *O. furnacalis* cell line, followed by immunoprecipitation (IP) and LC-MS/MS analysis, identified Ofznf-2 as an interacting partner for both OfMasc and OfMascNLS (Fig. 3a). When mCherry-tagged Ofznf-2-E or Ofznf-2-L was co-expressed with

**Fig. 2 | *Ofznf-2* is required for male-type splicing of *Ofdsx* and dosage compensation in *Ostrinia furnacalis*. a** Splicing patterns of *Ofdsx* mRNA in egg masses injected with *GFP* or *Ofznf-2* siRNA at 48 hpo. Similar results were obtained in three independent experiments. **b** Expression levels of *Ofdsx^M* in egg masses injected with *GFP* (n = 4) or *Ofznf-2* (n = 5) siRNA at 48 hpo. A two-sided Welch's *t*-test indicated a significant difference between the two groups, $t(5.1141) = 3.19$, $p = 0.024$, Cohen's $d = 2.24$, 95% CI = [0.138, 1.25]. **c** Chromosomal distribution of differentially expressed transcripts in egg masses injected with *Ofznf-2* siRNA at 48 hpo. The box

shows the interquartile range (IQR = Q3−Q1), with a line at the median. The whiskers extend from the edge of the box to the most extreme values within 1.5× IQR. Only values in the range −2 to 2 are shown. The numbers of transcripts analyzed per chromosome (ordered left to right in the panel) are: 675, 331, 607, 747, 759, 546, 486, 587, 539, 673, 380, 583, 578, 579, 410, 792, 539, 650, 545, 622, 469, 445, 642, 560, 231, 276, 387, 576, 387, 279, 343, and 632. **d** Sex ratio of adult moths emerging from embryos injected with *GFP-* or *Ofznf-2* siRNA, with sample sizes indicated for each group. Source data are provided as a Source data file.

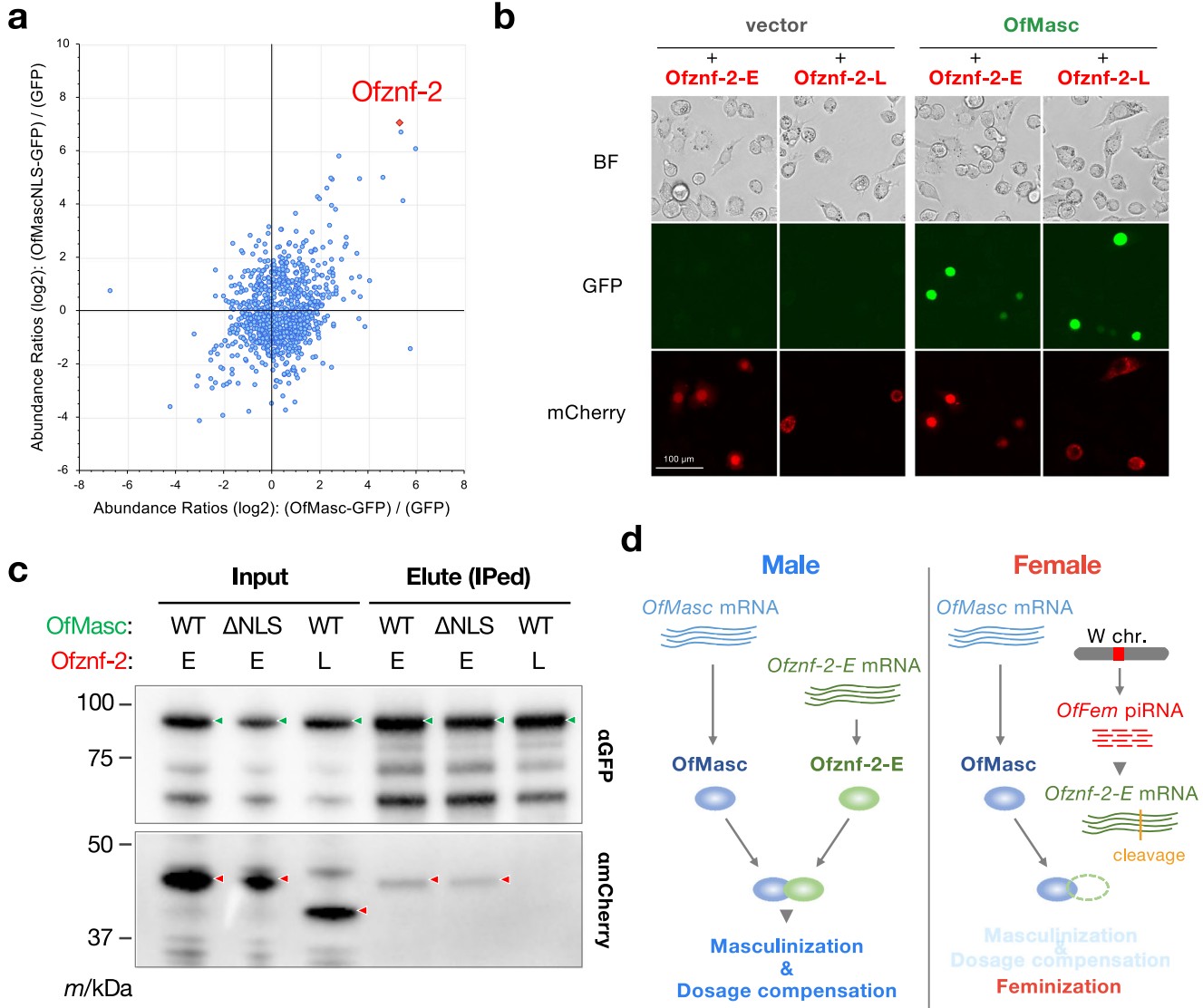

**Fig. 3 | Ofznf-2-E physically interacts with OfMasc. a** LC-MS/MS analysis of proteins co-immunoprecipitated with anti-GFP nanobody from cells transfected with *GFP*, *OfMasc-GFP*, or *OfMascNLS-GFP*. Dots represent *O. furnacalis* proteins quantified by label-free precursor ion analysis. **b** Fluorescence microscopy of BmN-4 cells co-transfected with *OfMasc-GFP* and *Ofznf-2-mCherry* cDNAs. Similar results were obtained in two independent experiments. **c** Co-immunoprecipitation experiments using BmN-4 cells co-transfected with *OfMasc-GFP* or *OfMascNLS-GFP*

and *Ofznf-2-mCherry* cDNAs. The immunoprecipitates with anti-GFP nanobody beads were immunoblotted using anti-mCherry antibody. GFP- and mCherry-fused protein bands are highlighted with green and red arrowheads, respectively. Similar results were obtained in two independent experiments. **d** Proposed model of the sex determination mechanism involving *Ofznf-2* and *OfMasc* in *O. furnacalis*. Source data are provided as a Source data file.

OfMasc-GFP or OfMascNLS-GFP in *B. mori* BmN-4 cells, only Ofznf-2-E-mCherry co-localized with OfMasc-GFP or OfMascNLS-GFP in the nucleus or cytoplasm (Figs. 3b, S9b and S13). A co-IP experiment further confirmed the protein interaction between OfMasc and Ofznf-2-E, but not between OfMasc and Ofznf-2-L (Fig. 3c). These results suggest that Ofznf-2-E, rather than Ofznf-2-L, physically associates with OfMasc to facilitate male-specific splicing of *Ofdsx* and dosage compensation

in males. In females, these processes are inhibited through *OfFem* piRNA-mediated cleavage of *Ofznf-2* mRNA (Fig. 3d).

**A *Wolbachia* hijacks the host feminizing piRNA function in *O. furnacalis***

Previous studies suggested that the endogenous feminizing factor is either absent or inactivated in the *Wolbachia*-infected lineage of

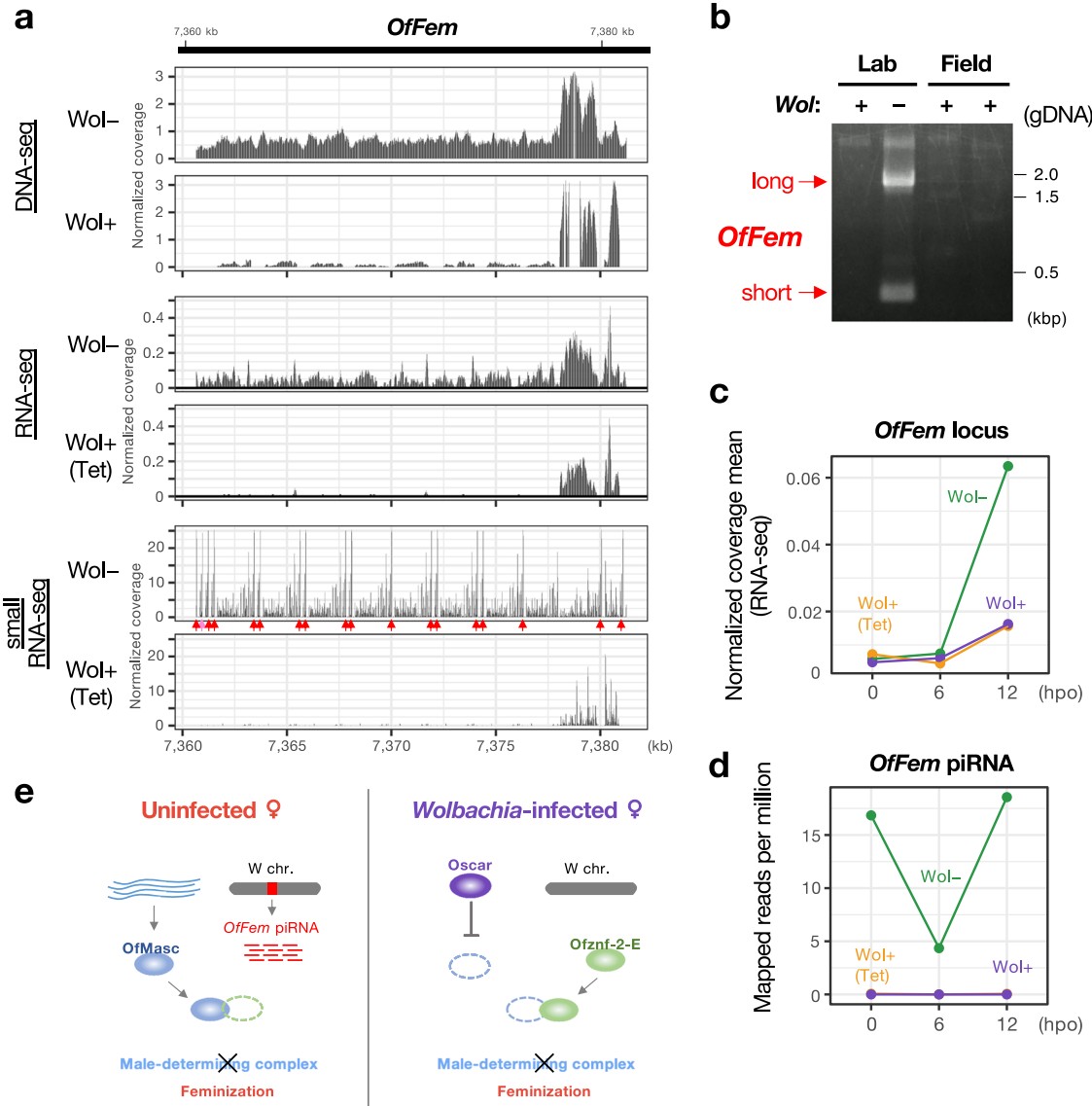

**Fig. 4 | *OfFem* piRNA is absent from the W chromosome of *Wolbachia*-infected *Ostrinia furnacalis*. a** Comparative mapping of DNA-seq, RNA-seq, and small RNA-seq onto the *OfFem* locus in *Wolbachia*-infected (Wol+), uninfected (Wol−), and tetracycline-treated *Wolbachia*-infected (Wol+ (Tet)) *O. furnacalis* egg masses at 12 hpo. Arrows indicate the positions of the *OfFem* piRNA-producing sites. Red arrows denote exact matches to *OfFem* piRNA, and a pink arrow shows a sequence with a single nucleotide mismatch. **b** PCR amplification of the *OfFem* locus from the genomes of field-collected *O. furnacalis* moths infected with male-killing *Wolbachia* (n = 2). Expression profile of the *OfFem* locus (**c**) and *OfFem* piRNA (**d**) in uninfected, *Wolbachia*-infected, and tetracycline-treated *Wolbachia*-infected egg masses. **e** Proposed model illustrating the sex determination system hijacked by male-killing *Wolbachia* in *O. furnacalis*. Source data are provided as a Source data file.

*Ostrinia* species[6–8]. In this lineage, the *Wolbachia* male-killing protein Oscar inactivates OfMasc protein, effectively substituting for the host feminizing factor[9,40]. Whole-genome sequencing with short reads from a *Wolbachia*-infected female revealed that almost no reads mapped to the *OfFem* locus (Fig. 4a). In the draft assembly of *Wolbachia*-infected *O. furnacalis*, the contig tig00004611 contained a single 222 bp-long region homologous to the *OfFem* piRNA-producing region (80.2% identity), but it lacked the repeat structure homologous to *Ofznf-2* (Fig. S14). Additionally, fragments of the *OfFem* locus could not be PCR-amplified from field-collected *Wolbachia*-infected *Ostrinia* individuals (Fig. 4b). Consistently, the expression levels of *OfFem* and *OfFem*-derived piRNA were nearly undetectable in the *Wolbachia*-infected lineage (Figs. 4a, c, d and S15). These findings indicate that the *OfFem* locus is absent from the W chromosome in the *Wolbachia*-infected lineage. As a result, this W chromosome lacks feminizing ability through the downregulation of *Ofznf-2* mRNA. This absence explains the sex-reversal phenomenon in *O. furnacalis* infected with

male-killing *Wolbachia*, where the depletion of *Wolbachia* from infected mothers leads to female killing (Fig. S16).

## Discussion

We recently identified the *Wolbachia* protein Oscar and demonstrated its feminizing function through the inhibition of OfMasc activity[9,40]. In this study, we identified the host feminizing factor *OfFem* and showed its absence in *Wolbachia*-infected females. These findings provide molecular evidence for sex-determination hijacking: *Wolbachia*'s Oscar has fully taken over the role of the host's feminizing piRNA in *O. furnacalis* during endosymbiosis (Fig. 4e). Beyond male killing in *Ostrinia* species, cytoplasmic sex determination, where an endosymbiont entirely hijacks the host's sex-determination factor, has been observed in two other female-heterogametic hosts: the terrestrial isopod *Armadillidium vulgare* (pill bug) and the butterfly *Eurema mandarina*, both infected with feminizing *Wolbachia*[3,4]. In *E. mandarina*, *Wolbachia* depletion results in all-male offspring, suggesting that

*Wolbachia* hijacks the role of the host's feminizing factor. Given that this feminizing *Wolbachia* encodes an *Oscar* homolog[48] and the W chromosome is entirely absent from infected strains of *E. mandariana*[4,5], it is likely that the loss of the feminizing piRNA and its replacement by *Oscar* has occurred in this host-endosymbiont system.

Most lepidopteran species infected with male-killing bacteria retain their original feminization system, as indicated by the restoration of a 1:1 sex ratio following antibiotic-induced bacterial depletion[35–39]. A recent study identified *Oscar* homologs in the genomes of male-killing *Wolbachia* infecting several lepidopteran hosts, supporting the feminizing activity of Oscar homologs[48]. This suggests that both *Oscar* and the host feminizing factor coexist in these host-endosymbiont combinations, and that the presence of redundant feminizing factors does not always lead to the endosymbiont's complete hijack of the host's sex determination system. On the other hand, in species where sex is determined by the number of Z chromosomes rather than a W chromosome-encoded feminizing factor[52], the feminizing system cannot be hijacked cytoplasmically. Considering the evolutionary lability of piRNA clusters[46,53], species with feminizing piRNA encoded on their W chromosomes may be more vulnerable to sex determination hijacking. Further investigations into the feminizing systems of lepidopteran insects infected with male-killing symbionts could help validate this hypothesis. Additionally, although a loss-of-function mutation in a W-linked feminizing factor is unlikely to provide any selective advantage, why has such a mutation become fixed in a Japanese population? We propose that founder effects are responsible for this. In *O. furnacalis*, nine *Wolbachia* strains have been identified in China[54], but only one of them (*w*Fur9) has been observed in Japan[33,34]. This suggests that after a small number of *w*Fur9-infected individuals carrying the *OfFem*-lacking W chromosome have been introduced from China to Japan, an increase in *Wolbachia* prevalence due to male killing has facilitated the rapid spread of the *OfFem*-lacking W chromosome, akin to the spread of neo-W chromosomes hitchhiking via male-killing *Spiroplasma*[55]. Investigating the presence of the *OfFem* locus in Chinese populations could clarify whether the *OfFem*-lacking W chromosome is unique to the Japanese population.

*Bombyx mori* has two homologs of *znf-2*, *Bmznf-2* (KWMTBOMO15133) and *Bmznf-2L* (KWMTBOMO15134), located tandemly on the 25th chromosome (Fig. S17a). In BmN-4 cells, both protein homologs displayed nuclear localization, and co-expression with BmMascNLS (a derivative of *B. mori* Masc lacking an active nuclear localization signal[56]) induced the nuclear localization of BmMascNLS, indicating their association with BmMasc (Fig. S17b–e). A co-IP experiment further confirmed the physical association of Bmznf-2 and Bmznf-2L with BmMasc (Fig. S17f, g). As previously reported[49], Bmznf-2 showed distinct masculinizing activity in BmN-4 cells (Fig. S17h), whereas Bmznf-2L exhibited a small positive effect (Fig. S17h). Notably, when co-expressed with BmMasc, both homologs enhanced BmMasc-dependent masculinizing activity (Fig. S17h). Together with evidence of the functional interaction between OfMasc and Ofznf-2, these findings suggest that a complex comprising two or more CCCH-tandem zinc finger proteins is likely a conserved regulatory element for male-determining functions in Lepidoptera. Recently, van't Hof et al. proposed that a heterozygotic combination of *Masc* alleles is essential for male development in *Bicyclus anynana* butterflies[29]. This strongly supports the hypothesis that heterodimerization of CCCH-tandem zinc finger proteins is crucial for forming a functional male-determining complex, with the composition of CCCH-tandem zinc finger proteins in this complex potentially varying across species.

In this study, we identified *OfFem*, the feminizing factor on the W chromosome, in uninfected *O. furnacalis*. This marks the fourth instance of a primary sex determiner being molecularly identified in Lepidoptera[18,19,30]. *OfFem* acts as a precursor to a single female-specific piRNA, which targets *Ofznf-2* mRNA. Functional *Ofznf-2* genes were identified at multiple genomic loci and categorized into two types: *Ofznf-2-E* and *Ofznf-2-L*. While *Ofznf-2-E* is expressed early in embryogenesis, *Ofznf-2-L* is expressed throughout embryogenesis. Knockdown experiments revealed that either or both *Ofznf-2* types are essential for two male-specific biological processes: male-type *dsx* splicing and dosage compensation. In two other lepidopteran species, *B. mori* and *P. xylostella*, the male-specific processes are inactivated by feminizing piRNAs targeting *Masc* mRNA[18,19]. The homologs of *Masc* in these species perform conserved roles in these processes. Therefore, all feminizing factors identified so far are female-specific piRNAs that target genes essential for male-determining processes. Given the physical interaction between OfMasc and Ofznf-2-E in *O. furnacalis*, other genes encoding components of the male-determining complex are plausible targets for W-linked feminizing piRNAs. Future studies may uncover the diversity of feminizing piRNA targets, emphasizing the role of the repetitive W chromosome as the origin of feminizing piRNAs. Additionally, we found that *OfFem* piRNA is derived from the tandem, degenerate copies of *Ofznf-2* on the W chromosome. Similarly, feminizing piRNAs in *P. xylostella* originate from the retrocopies of *Masc* on the W chromosome[19]. These findings strongly suggest that the transposition of genes with male-determining functions to the W chromosomes represents the initial step in the evolution of feminizing piRNAs.

## Methods

### Insect rearing and tissue collection
The founder *O. furnacalis* moths of the laboratory strain were collected during the early summer seasons of 2014–2021 in Japan. The laboratory strains were maintained according to previously described methods[32]. Egg masses were harvested within a 2-h time window at designated time points for further experiments. Genomic DNA and total RNA were extracted using the DNeasy Blood & Tissue Kit (QIAGEN) and TRI reagent (Molecular Research Center), respectively.

### Genomic PCR and reverse transcription-PCR (RT-PCR)
The *OfFem* locus was amplified using KOD One DNA polymerase (TOYOBO, Japan) with the primers designed to match the homologous repeats to *Ofznf-2* in the *OfFem* locus. The sex-specific splicing pattern of *Ofdsx* was examined by RT-PCR using KOD FX−neo DNA polymerase (TOYOBO, Japan) as previously described[32]. Quantitative RT-PCR (RT-qPCR) was performed with the KAPA SYBR FAST qPCR kit (Kapa Biosystems). Expression levels of $Ofdsx^M$ were normalized to $Ofdsx^C$, the common region of the male- and female-type *Ofdsx*. RT-qPCR primers specifically targeting *Ofznf-2-E* are detailed in Fig. S6, while all primers used in this study are listed in Table S3.

### Modified rapid amplification of cDNA ends (RACE)
Modified RACE experiments followed a previously established protocol with minor adjustments[18,57]. The GeneRacer Kit (Invitrogen) was employed according to the manufacturer's instructions, excluding calf intestine phosphatase (CIP) treatment. Omitting this step enabled adapter oligonucleotide ligation to both the capped and 5′-OH mRNA molecules for comprehensive 5′-end analysis.

### RNA-seq and small RNA-seq
Total RNA samples were isolated from individual eggs at 18, 24, and 48 hpo and were molecularly sexed following established protocols[32]. Sexed egg samples were either pooled by sex or analyzed individually. RNA-seq libraries were constructed from these samples using the Stranded mRNA Prep, Ligation Kit (Illumina), and indexed with IDT for Illumina RNA UD Indexes SetA. The libraries were sequenced on the NovaSeq 6000 Sequencing System (Illumina, San Diego, CA, USA) with 100-bp paired-end reads. Small RNA-seq was performed on samples depending on the experimental setup. For sex-pooled egg samples, RNA fractions of 18−30 nucleotides in length were isolated and used to

construct small RNA-seq libraries. Sequencing of these libraries was conducted on the DNBSEQ platform, with library preparation and sequencing services provided by BGI (Shenzhen, China). For gonad and egg mass samples, total RNA was isolated from the testes of eight males, the ovaries of two to seven females, and egg masses collected from five females at each time point. Small RNA fractions (20–30 nucleotides in length) were isolated for these samples as described previously[32]. Small RNA-seq libraries were constructed and sequenced using the Illumina HiSeq 2500 platform. For RNA-seq analysis of egg mass samples, RNA-seq libraries were constructed using the Ribo-Zero rRNA Removal Epidemiology Kit and Illumina TruSeq RNA Prep Kit, which are non-strand-specific and omit poly(A) selection. These libraries were sequenced on the Illumina HiSeq 2500 platform.

### Identification and expression analysis of *OfFem* locus and *OfFem* piRNA

The de novo transcriptome assembly was performed using the Trinity software package[58]. Contig expression levels were quantified using Salmon[59], and the output files were processed with the R package Wasabi (https://github.com/COMBINE-lab/wasabi). Sex-biased contigs with statistically significant expression differences between male and female samples ($q$ value < 0.05) were identified using the Sleuth software tool[60]. Two contigs that met several criteria (see Results) were mapped on the single *OfFem* locus identified from an in-house assembled whole-genome assembly (version #2), which is used for further analysis.

RNA-seq data were mapped to the *OfFem* locus using HISAT2[61]. Coverage depth at the locus was calculated with SAMtools[62] and BEDTools[63]. Small RNA-seq data were mapped to the *OfFem* locus using Bowtie[64], allowing up to two nucleotide mismatches and permitting multi-maps. *OfFem* piRNA reads were specifically quantified by mapping small RNA-seq data onto a single representative repeat of the *Ofznf-2* sequence within the *OfFem* locus using BWA[65] with no mismatches allowed. To infer the structure of the repeat elements in the *OfFem* locus, a custom repeat library was constructed de novo from the *O. furnacalis* whole-genome sequence (version #2) using RepeatModeler2[66]. The genome was subsequently annotated with RepeatMasker[67], employing this custom repeat library. Homologous repeats to the *Ofznf-2* sequence within the *OfFem* locus were identified by performing a BLASTn search with the nucleotide sequence of *Ofznf-2-L* as the query.

### Phylogenetic analysis

Homologs of Ofznf-2, limited to the top hit from each species, were identified through a BLASTp search against the NCBI non-redundant protein database (Table S4). BLASTp search was performed via the NCBI BLAST homepage (https://blast.ncbi.nlm.nih.gov/) with default parameters, using as the query the partial protein sequence of Ofznf-2 ("MCKDWVRGTCARGAACIYAHELDKDQLKGVYRFCRDFENDRCERQVC YFVHATTFEKEHFFRTAFLPPHALHHLKT"). Phylogenetic analysis of Ofznf-2 homologs was conducted alongside previously published Masc protein sequences (Table S4). Amino acid sequences were aligned using MAFFT[68] with default settings, and the alignments were trimmed with trimAl[69] in "automated1" mode. A maximum likelihood tree was constructed via the IQ-TREE web server[70], with node support calculated from 1000 ultrafast bootstrap replicates[71].

### Analysis of dosage compensation

RNA-seq libraries were prepared from total RNA using the Illumina platform protocols and sequenced as stranded libraries on the Nova-Seq 6000 Sequencing System (Illumina, San Diego, CA, USA) with 150-bp paired-end reads. Library preparation and sequencing were performed by Novogene (Beijing, China). RNA-seq data were mapped to the annotated *O. furnacalis* genome (version #2) using HISAT2[61]. Gene expression levels were quantified with featureCounts[72]. Counts per million values and fold-change metrics were calculated using the

edgeR package, employing normalization by the trimmed mean of M-values method[73].

### Cell culture and transfection

The DNA fragments encoding *OfMasc-GFP*, *Ofznf-2*, and their derivatives were cloned into the pIZ/V5-His-g3 vector[51]. *B. mori* BmN-4 cells were cultured at 26 °C in IPL-41 medium (Applichem, Germany) supplemented with 10% fetal bovine serum (FBS; Gibco, USA). BmN-4 cells ($4.0 \times 10^5$ cells per 35-mm diameter dish) were transfected with two plasmid DNAs (1 µg each) using FuGENE HD transfection reagent (Promega, USA). The localization of GFP-fused and mCherry-fused proteins was examined at 72 h post-transfection using either a FLoid™ cell imaging station or an EVOS M5000 cell imaging station (Thermo Fisher Scientific). *O. furnacalis* OfT1C/tet cells, derived from *Wolbachia*-infected *O. furnacalis* embryos[9], were cultured at 26 °C in Express Five™ SFM medium (Gibco, USA) supplemented with 18 mM L-glutamine, 10% FBS, and 3 µg/mL tetracycline. We isolated three RT-PCR clones of *Ofznf-2-E* from *O. furnacalis* embryonic cDNA. These clones encoded proteins with slight differences in amino acid sequences, but all displayed similar localization patterns in BmN-4 cells (Fig. S9). Of these, we used *Ofznf-2-E*_clone #1 as the representative clone for all subsequent cell-based experiments.

### LC-MS/MS-based identification of OfMasc-interacting proteins

OfMasc-GFP, OfMascNLS-GFP, or GFP were transiently expressed in OfT1C/tet cells ($3.2 \times 10^6$ cells) seeded in 10 cm-diameter culture dishes by transfection using FuGene HD (Promega). OfMasc-interacting proteins were co-immunoprecipitated with OfMasc-GFP using GFP-Trap magnetic agarose beads (ChromoTek, Germany). Proteins bound to the beads were digested by Trypsin/Lys-C mix (Promega). LC-MS/MS analysis of the resultant peptides was performed using an EASY-nLC 1200 UHPLC connected to an Orbitrap Fusion mass spectrometer equipped with a nanoelectrospray ion source (Thermo Fisher Scientific). The peptides were separated on a 75 µm inner diameter × 150 mm C18 reversed-phase column (Nikkyo Technos, Japan) with a linear 4–32% acetonitrile gradient for 0–100 min followed by an increase to 80% acetonitrile for 10 min. The mass spectrometer was operated in a data-dependent acquisition mode with a maximum duty cycle of 3 s. MS1 spectra were measured with a resolution of 120,000, an automatic gain control (AGC) target of 4e5, and a mass range from 375 to 1500 *m/z*. Higher-energy collisional dissociation MS/MS spectra were acquired in the linear ion trap with an AGC target of 1e4, an isolation window of 1.6 *m/z*, a maximum injection time of 35 ms, and a normalized collision energy of 30. Dynamic exclusion was set to 20 s. The mass spectrometry experiments were performed once for each sample. The raw LC-MS/MS data were directly analyzed against *O. furnacalis* protein data (GCF_004193835.1_ASM419383v1_protein.faa) downloaded from NCBI, supplemented with sequences for OfMasc-GFP and GFP-Trap. Analyses were performed using Proteome Discoverer version 2.5 software (Thermo Fisher Scientific) with Sequest HT search engine. The search parameters were as follows: (a) trypsin as an enzyme with up to two missed cleavages; (b) precursor mass tolerance of 10 ppm; (c) fragment mass tolerance of 0.6 Da; (d) carbamidomethylation of cysteine as a fixed modification; and (e) acetylation of protein N-terminus and oxidation of methionine as variable modifications. Peptides and proteins were filtered at a false discovery rate (FDR) of 1% using the percolator node and the protein FDR validator node, respectively. Label-free precursor ion quantification was performed using the precursor ions quantifier node, and normalization was performed such that the total sum of abundance values for each sample over all peptides was the same.

### Western blotting and immunoprecipitation

Western blotting and immunoprecipitation experiments were performed as previously described[9]. Lysates from BmN-4 cells expressing

GFP-tagged proteins were incubated with GFP-Trap magnetic agarose beads. After binding, the beads were washed three times and boiled to elute the bound proteins. The eluted proteins were separated by SDS-PAGE and transferred onto PVDF membranes. For immunodetection, an anti-GFP antibody (1:5000 dilution, MBL, 598, Japan) or an anti-RFP antibody (1:5000 dilution, MBL, PM005, Japan) was used as the primary antibody.

## Microinjection of siRNA

Microinjection of siRNA was carried out as previously described[32]. Specifically, 100 μM siRNA solutions were injected into *O. furnacalis* embryos within 2 hpo. For Figs. 2d and S12, however, 50 μM siRNA was used. Egg masses injected with siRNA were collected at 48 hpo for analysis or allowed to hatch. Hatched larvae were raised to the pupal stage and sexed based on pupal terminal abdominal morphology. siRNAs were purchased from FASMAC Corp. (Atsugi, Japan), prepared in annealing buffer (100 mM potassium acetate, 2 mM magnesium acetate, 30 mM HEPES-KOH, pH 7.4) at concentrations of 100–500 μM, and stored at −80 °C. Nucleotide sequences for the *Ofznf-2* siRNA used in this study are listed in Table S3.

## DNA extraction and genome sequencing

Whole-genome shotgun sequencing was performed using PacBio and Illumina sequencing platforms. Genomic DNA was extracted from female larvae of laboratory strains (T1-A and T1-C1) of *O. furnacalis* using the Blood & Cell Culture DNA Midi Kit and a Genomic-tip Kit, respectively (QIAGEN, Hilden, Germany). DNA quality and concentration were assessed using a Qubit 4 Fluorometer (Thermo Fisher Scientific, MA, USA) and a Pippin Pulse electrophoresis system (Agilent Technologies, CA, USA).

For PacBio sequencing, a continuous long read (CLR) library was constructed from the T1-A sample using the SMRTbell Express Template Prep Kit 2.0 (Pacific Biosciences, CA, USA), followed by size selection with the Blue Pippin system (Saga Science, MA, USA) using a 30 kb cutoff. The library was sequenced on the PacBio Sequel II system with a Binding Kit 2.0 and Sequencing Kit 2.0 (Pacific Biosciences, CA, USA) using 30-h movies. Raw sequencing data from a single 8M SMRT cell were processed using the PacBio SMRT Link v11.0.0.144466 software. For Illumina sequencing, a paired-end library was generated using the Illumina DNA PCR-Free Library Prep Tagmentation Kit (Illumina, CA, USA) and sequenced on the NovaSeq 6000 platform (Illumina, CA, USA) with a 2 × 150 bp read length.

For High Fidelity (HiFi) sequencing, genomic DNA from the T1-C1 sample was fragmented into 15–20 kb sizes using a g-tube device (Covaris Inc., MA, USA). A HiFi library was prepared with the SMRTbell Prep Kit 3.0 (Pacific Biosciences, CA, USA) and subsequently size-selected with 35% AMPure PB beads (Pacific Biosciences, CA, USA). Sequencing was conducted on the PacBio Sequel II system using the Binding Kit 3.2 and Sequencing Kit 2.0 (Pacific Biosciences, CA, USA) with 30-h movies. HiFi reads were generated and processed from one 8M SMRT cell using the DeepConsensus v1.2 program under the default parameters[74]. Genomic DNA was also fragmented to an average size of 500 bp with the focused-ultrasonicator M220 (Covaris Inc., MA, USA). A paired-end library was prepared with the TruSeq DNA PCR-Free Library Prep Kit (Illumina, CA, USA) and size-selected using an agarose gel and the Zymoclean Large Fragment DNA Recovery Kit (Zymo Research, CA, USA). Sequencing library was then conducted on the NovaSeq 6000 system (Illumina, CA, USA) with a read length of 2 × 150 bp.

## Hi-C library preparation and sequencing

The Hi-C library, derived from the T1-C8 larval body sample, was prepared following the Omni-C Kit protocol (Dovetail, CA, USA). Briefly, chromatin was fixed in the nucleus using disuccinimidyl glutarate and formaldehyde, then digested with DNase I. Extracted chromatin DNA fragments were end-repaired and ligated to a biotinylated bridge adapter. After proximity ligation, crosslinks were reversed, and DNA was purified. Sequencing library construction involved Illumina-compatible adapters, with biotin-labeled fragments enriched via streptavidin magnetic beads and PCR amplification. The final library was sequenced on the Illumina NovaSeq 6000 system with 2 × 150 bp paired-end reads, yielding a total of 98.3 Gb of sequence data.

## Chromosome-level genome assembly

Two versions of the *O. furnacalis* genome assembly (version #1 and version #2) were constructed using PacBio CLR reads and HiFi reads, respectively. For version #1, PacBio CLR reads were assembled with Canu v2.1.1[75], with the parameters "genome_size=400m corOutCoverage=200 batOptions="-dg 3 -db 3 -dr 1 -ca 500 -cp 50 " -pacbio-raw." The resulting contigs were polished with Pilon v1.4.0[76] using Illumina reads. To address redundancy caused by the independent assembly of haplotypes, haplotigs were removed with Purge_Dups v1.2.5[77]. Hi-C scaffolding was then performed using an Omni-C dataset to produce a chromosome-level assembly.

Before Hi-C scaffolding, 11 W chromosome-derived contigs, initially removed by the Purge_Dups process, were rescued using Hi-C contact information. Hi-C scaffolding was conducted with YaHS v1.2a[78], utilizing Omni-C read mapping information generated by HiLine v0.2.4 (https://github.com/wtsi-hpag/HiLine). The Hi-C contact map was visualized with Juicebox v2.16.00[79], underwent detailed manual curation to resolve assembly errors and mis-scaffoldings.

For version #2, the genome assembly was generated using Hifiasm v0.19.6[80] with the parameter "--primary -s 0.30," based on PacBio HiFi reads. The "-s" parameter was optimized by comparing assembly statistics from several trial runs, with the primary goal of reducing redundancy caused by unresolved haplotypes in the primary assembly. Obtained contigs were polished with NextPolish2 v0.2.0[81] using PacBio HiFi reads and Illumina reads. Subsequent Hi-C scaffolding and manual curation were performed on the primary contigs (p_ctg) using the same procedures as for version #1.

Both assemblies demonstrated robust results, with N50 values greater than 16 Mbp and demonstrating >99% completeness of conserved genes, as evaluated by BUSCO[82]. Detailed assembly statistics for both versions are summarized in Table S1. The chromosome numbers were assigned based on synteny with the chromosomes of *B. mori* (Table S5).

## Statistics and reproducibility

All statistical tests were two-sided. The significance threshold and tests used are indicated in each figure legend. Experiments were independently replicated as follows: Fig. 2a, three independent biological replicates; Fig. 3b, c, two independent biological replicates; Fig. 1f, two technical replicates; Figs. 1d (n = 3) and 4b (n = 2), one technical replicate.

## Reporting summary

Further information on research design is available in the Nature Portfolio Reporting Summary linked to this article.

# Data availability

The genome sequences of *O. furnacalis* have been deposited in the DNA Data Bank of Japan under accession numbers AP039452–AP039489 (version #1) and AP039490–AP039545 (version #2). Specific accession numbers for each chromosome can be found in Supplementary Data 1. Associated data, including raw sequencing reads, are available under PRJDB14515. All the other raw sequencing data have been deposited in GenBank under accession numbers DRR462151–DRR462186 [https://ddbj.nig.ac.jp/search/entry/sra-study/DRP014454] (RNA-seq), DRR629067–DRR629078 [https://ddbj.nig.ac.jp/search/entry/sra-study/

DRP014461], DRR632127–DRR632132 [https://ddbj.nig.ac.jp/search/entry/sra-study/DRP014456] (RNA-seq), DRR623824–DRR623829 [https://ddbj.nig.ac.jp/search/entry/sra-study/DRP014460], DRR408740–DRR408744 [https://ddbj.nig.ac.jp/search/entry/sra-submission/DRA014893], DRR626685–DRR626691 [https://ddbj.nig.ac.jp/search/entry/sra-study/DRP009884] (small RNA-seq), SRR31936107 (wFur-infected genome, long read), SRR31936108 (wFur-infected genome, short read), and SRR31944223 (wFur-uninfected genome, short read). Accession numbers for publicly available data used in phylogenetic analysis can be found in Supplementary Table 4. The MS proteomics data have been deposited to the ProteomeXchange Consortium via the jPOST partner repository with the dataset identifier PXD059481 [https://repository.jpostdb.org/entry/JPST003537]. Source data are provided with this paper.

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

## Acknowledgements

The authors thank M. Kawamoto and K. Nishino for their contribution to the initial stage of the project, the Technology Advancement Center, Graduate School of Agricultural and Life Sciences, The University of Tokyo, for TEM analysis, the Institute for Sustainable Agro-ecosystem Services, The University of Tokyo, for insect collection, and C. Yasunaga-Aoki for providing the BmN-4 cell line. This work was supported by Grants-in-Aid for Scientific Research (JP17H06431 to S.K. and T.Ki.; JP22H00366 to S.K.; JP24H02289 to S.K.; JP24H02278 to T.Ki.; JP21J12325 to T. Fukui; P23KJ0468 to T.M.; JP22H04925 (PAGS) to Y.S., A.T., and T.I.; JP24KJ0036 to T. Fukui) and G-7 Scholarship Foundation (no grant number) to S.K.

## Author contributions

T. Fukui, T.M., and S.K. conceived and designed the experiments; T. Fukui, N.M.-I., T.Ka., T. Fujii, T.Ki., and S.K. performed molecular biological and biochemical experiments; H.K. performed LC-MS/MS analysis; T. Fukui, T.M., H.H., K.S., Y.S., A.T., and T.I. performed bioinformatic analysis; T. Fukui, T.M., N.M.-I., K.S., T.Ki., and S.K. discussed the results; T. Fukui and S.K. wrote the manuscript with intellectual input from all authors. S.K. supervised the project.

## Competing interests

The authors declare no competing interests.
