## [Transparent Peer Review file · Nature Communications]

Complete transition from chromosomal to cytoplasmic sex determination during prolonged Wolbachia symbiosis

Corresponding Author: Dr Susumu Katsuma

Version 0:

Reviewer comments:

Reviewer #1

(Remarks to the Author)

In a previous study, Susumu Katsuma's team identified a protein called Oscar, through which the endosymbiotic Wolbachia inhibits the Masculinizer (OfMasc) protein in *Ostrinia furnacalis*, thus explaining the molecular mechanism of Wolbachia-induced male killing, which probably also applies to other lepidopteran species (Katsuma et al. 2022). In this study, S. Katsuma's team answered another fundamental question by showing that Wolbachia completely takes over the feminizing function in the Wolbachia-infected lineage of *O. furnacalis*, which explains why the tetracycline-cured Wolbachia male-killing strains of *O. furnacalis* produce only male offspring while the females die. Furthermore, the authors discovered that in a Wolbachia-free *O. furnacalis* population, feminization is controlled by the W chromosome-derived OfFem piRNA, which surprisingly does not target the OfMasc mRNA but the mRNA of the Ofznf-2 gene and inhibits the OfMasc/Ofznf-2 masculinizing protein complex. Thus, there are two sex-determining pathways in *O. furnacalis*, one with the dominant role of the W chromosome, controlled by the feminizing piRNA produced by the W-linked gene, and the other in the Wolbachia-infected lineage, where feminization is taken over by Wolbachia and the W chromosome has lost the feminizing gene and plays no role in sex determination. Overall, this is a fascinating story that deserves to be published in this highly respected journal.

The essential role of the Ofznf-2 gene in sex determination is an important part of this study and the name of this gene should therefore appear in the abstract. Below are my comments and minor suggestions for you to consider when making some revisions to the text.

Specific comments

(1) Results, lines 88-89: The timing of sex determination in embryos of *Ostrinia furnacalis*, which is given here and in several other places in the results as "during the period of sex determination", is referred to Fukui et al. (2023). As this is important for a better understanding of your results, could you define this period? Do you mean the period from the first appearance of the male-specific splice variant of Ofdsx (12 h) to the completion of the feminization and masculinization processes (48 h)?

(2) Fig. S3a: Strains D, E, F and α , β , γ are not mentioned in the Methods and should therefore be explained in the figure legend. What is the difference between the strains (D, E, F) and (α , β , γ)? Yes, fragments of the OfFem locus were amplified in all female samples, but especially in the (α , β , γ) strains, several bands can also be seen in the male DNA samples. Make this clear.

(3) Table S2: The Ofznf-2 copies identified in chromosome 24 in the genome sequenced by Dai et al. (2024) have the same length as those in genome sequence version 2, where the copies, however, are located on chromosome 28. It is obvious that they are the same copies, but that there is chaos in the chromosome numbering. For *O. furnacalis*, did you use the numbering according to the silkworm where chr 28 matches chr 26 and chr 24 matches chr 28+31 in the *O. furnacalis* assembly of Dai et al. (2024)? See Fig. 2B in Dai et al. (2024). Is it possible that chr 28 in the version 2 genome sequence is identical to chr 24 in the Dai et al. (2024) assembly? The same applies to Fig. S8. Make this clear.

(4) References in Supplementary Tables 2, 3, 4: Some of the references given in these tables are not used in the main text and therefore not listed in the References of the main text. I recommend listing each reference used in these tables under the corresponding supplementary table.

(5) Fig. 4a legend (lines 779-780): I think that Wol- should stand for Wolbachia-uninfected and Wol+ for Wolbachia-infected. Wol+(Tet), which stands for tetracycline-treated Wolbachia-infected, should also be explained, as well as the red (and one pink) arrows.

(6) Discussion, lines 195-198: Please check references 6, 7 and 8 for the Wolbachia sex-determining role in the terrestrial isopod *Armadillidium vulgare* and the butterfly *Eurema mandarina*. The references given here refer to publications on *O. furnacalis* and *O. scapularis*.

(7) Discussion, lines 243-245: I believe that this is not the third, but the fourth case of a primary sex determiner that has been molecularly identified in Lepidoptera: Fem in *Bombyx mori*, PxyFem in *Plutella xylostella*, Fet-W in *Lymantria dispar* (see Moronuki et al. 2025), and OfFem in *O. furnacalis*.

(8) Methods, line 357: Please check the reference to “Wolbachia-infected *O. furnacalis* embryos”. You refer to No. 11 (Harumoto and Lemaitre, 2018), but this publication is about *Spiroplasma* in *Drosophila*.

(9) Methods, lines 366 and 378: Please check the reference to “LC-MS/MS experiments”. You refer to No. 11 (Harumoto and Lemaitre, 2018), but the LC-MS/MS analysis is not described there. The same applies to this reference to “Western blotting and immunoprecipitation”.

(10) References, lines 712-716: No. 83 is not given in the text, but “DeepConsensus” is mentioned in line 422 and referred to No. 74. Please check it.

Minor suggestions

Fig. S2: Although the meaning of “Wol-” and “Wol+(Tet)” can be deduced, it is better to explain it in the figure legend. The same applies to Fig. S4.

Fig. S5: “the” before “OfFem pi RNA” should be written in normal font.

Fig. S7(b, c) legend: expression

Fig. S8 legend: reference (41)

L129: I suggest using “orthologs” instead of “homologs” because there are no data on the function of most znf-2 proteins in the species given in this phylogenetic cladogram.

Fig. S9, Fig. S13, and Fig. S17 legends: Correct “EGFP-“ to “GFP-”.

Fig. S15: (a) Mapping of small RNA-seq reads ... (b) Mapping of RNA-seq reads ...

L281: Fig. S6 [in this line, you refer to fig. S5, but this figure does not show positions of the primers]

L282 and L396: Table S3

L384-386: The text starting with “The Methods section ...” should be removed.

L465: Table S1

L496: Biol. Lett.

L570: Correct the publication year 2024 to 2025 in Moronuki et al.

L774: In Fig. 3c, the green and red dots, highlighting the GFP- and mCherry-fused protein bands, look more like arrowheads or triangles than dots.

František Marec
29 March 2025

Reviewer #2

(Remarks to the Author)

The manuscript titled “Complete transition from chromosomal to cytoplasmic sex determination during prolonged Wolbachia symbiosis” by Fukui et al is reviewed. This study comprehensively investigates the molecular mechanism of sex determination of *Ostrinia furnacalis* with and without infection of Wolbachia and provides strong evidence that male-killing Wolbachia hijacks the feminizing piRNA function in *Ostrinia furnacalis*. I appreciate the amount of effort in this work and its achievement and only have some minor suggestions to the manuscript.

Line 63: This is optional since it has been mentioned in the discussion but I was wondering if such sex determination in the

wild could cause any intraspecific diversification. So I think having information about the infection rate here in the introduction may provide clear picture about how important such sex determination mechanisms are to the evolution of the species.

Line 182: I find this figure (S16) very helpful in summarizing the whole picture, I think it would be better to have this figure in the main paper but not supplementary.

Line 214-216: Here comes inference about genetic drift, I guess it is related to the next section about the prevalence but I think it would be better to have more context about how the fixation of the mutation to the evolutionary history of the population in Japan.

Line 311-318: I was a bit confused here as I thought OfFem was identified based on the expression difference, but how the “expression analysis of the OfFem locus” section comes after the identification of OfFem? I wonder if merging these two sections could make it easier for readers to understand the flow of the work.

Line 333: was there blast criterion applied in blastp? If so, please specify.

Line 400: were these samples known for their sex? Are they from the lab breeding lines?

Line 458: I wonder if it is required to lower “-s” here, I assume the samples are quite inbred and should have low heterozygosity? If now, maybe it will be better to provide the evaluated heterozygosity and to make sense the “-s” parameter change.

Reviewer #3

(Remarks to the Author)

This study reports fascinating research that will be of wide interest with respect to the evolution of insect sex determination systems and of symbiont reproductive manipulations. The authors achieve two objectives; firstly, they identify a new primary Lepidopteran feminising factor in *O. furnacalis* ‘OfFem’, a W-linked piRNA that silences a constitutively expressed masculinising factor, and secondly, they demonstrate that the OfFem locus has been lost in a *O. furnacalis* lineage carrying the male-killing *Wolbachia* strain wFur. These are important, novel contributions, explaining convincingly how ‘cytoplasmic sex determination’ has arisen in this species. The research is of a very high standard, robustly conducted and the results convincing. The data are clearly presented, the manuscript is well written and is for the most part easy to follow, with an excellent discussion.

It would be useful to orient the reader to provide in the introduction a short overview of how widespread is the wFur strain in *O. furnacalis* in Japan, and where it occurs what is the typical prevalence. Are there any wFur lineages that have produced 1:1 sex ratios on tet treatment to remove *Wolbachia*?

Specific comments:

Line 108: states that there are 15 repeats of the Ofzfnf-2 target, but Fig. S5 seems to suggest that there are 17 repeats. Please clarify.

L214 ‘it must overcome random genetic drift’ - could there be a selective advantage associated with not expressing the W-linked feminizing factor in the presence of *Wolbachia*?

Fig. 3b

It is difficult to gauge the level of interaction between OfMasc and the Ofzfnf-2 variants from these images. Firstly, since the interpretation depends on co-localisation, a ‘merge’ panel for the fluorescent images could be provided. In addition, the images of cytoplasmic/nuclear co-localisation of Ofzfnf-2-E with OfMasc and OfMascNLS in Fig S13 are more convincing of interaction than the images in Fig 3b. and thus Fig 3b could be replaced with Fig S13 in the main text – especially if images can be added showing that Ofzfnf-2-L does not show similar nuclear/cytoplasmic co-localisation.

Fig. S3

what the different strains in panel A refer to – there does not seem to be an indication of what they represent anywhere in the ms.

Fig. S4C

Since the presence of OfFem piRNAs is dependent on the host lineage rather than the *Wolbachia* infection status per se, rather than using Wol-, Wol+(Tet) designations it would be clearer to replace with ‘Wol- lineage’, ‘Wol+ lineage (tet-treated)’. Similarly for the Figure legend of C, emphasise host lineage: e.g. ‘in the ovary of tetracycline-treated moths from the *Wolbachia*-infected lineage’.

Minor / Typos

Fig. S7 legend: ‘expression’ missing e.

Line 255: misspelling of *O. furnacalis*

227 ‘both protein homologs’ would be clearer

232 exhibited a little positive effect – remove little or replace with small

Reviewer #4

(Remarks to the Author)

Version 1:

Reviewer comments:

Reviewer #1

(Remarks to the Author)

Well done! In the revised manuscript, the authors have satisfactorily addressed all my comments and suggestions. I look forward to the publication of this excellent study.

Note: I have found two minor problems that can be fixed in the further editing of this article. These are:

(1) Fig. 4a legend (line 796): The explanations for Wol-, Wol+ and Wol+(Tet) were added as suggested in my comment #5, but the explanations for Wol- and Wol+ were incorrectly switched [see "Wolbachia-infected (Wol-), uninfected (Wol+)"]. I believe that Wol- should stand for Wolbachia-uninfected and Wol+ for Wolbachia-infected.

(2) L255-256: At the end of this sentence there could be references to the three previously identified primary sex determinants to clarify it as follows "This marks the fourth instance of a primary sex determiner being molecularly identified in Lepidoptera (18, 19, 30)."

František Marec
10 August 2025

Reviewer #2

(Remarks to the Author)

My comments have been adequately addressed in the responses, and the corresponding revisions have been made in the revised manuscript.

Point-by-point responses to the reviewer's comments

Reviewer #1:

In a previous study, Susumu Katsuma's team identified a protein called Oscar, through which the endosymbiotic *Wolbachia* inhibits the Masculinizer (OfMasc) protein in *Ostrinia furnacalis*, thus explaining the molecular mechanism of *Wolbachia*-induced male killing, which probably also applies to other lepidopteran species (Katsuma et al. 2022). In this study, S. Katsuma's team answered another fundamental question by showing that *Wolbachia* completely takes over the feminizing function in the *Wolbachia*-infected lineage of *O. furnacalis*, which explains why the tetracycline-cured *Wolbachia* male-killing strains of *O. furnacalis* produce only male offspring while the females die. Furthermore, the authors discovered that in a *Wolbachia*-free *O. furnacalis* population, feminization is controlled by the W chromosome-derived *OfFem* piRNA, which surprisingly does not target the *OfMasc* mRNA but the mRNA of the *Ofzmf-2* gene and inhibits the OfMasc/Ofzmf-2 masculinizing protein complex. Thus, there are two sex-determining pathways in *O. furnacalis*, one with the dominant role of the W chromosome, controlled by the feminizing piRNA produced by the W-linked gene, and the other in the *Wolbachia*-infected lineage, where feminization is taken over by *Wolbachia* and the W chromosome has lost the feminizing gene and plays no role in sex determination. Overall, this is a fascinating story that deserves to be published in this highly respected journal.

(Authors' reply)

We thank Prof. František Marec for reviewing this manuscript and giving encouraging comments.

The essential role of the *Ofzmf-2* gene in sex determination is an important part of this study and the name of this gene should therefore appear in the abstract.

(Authors' reply)

As suggested by this Reviewer, we added the gene name *Ofzmf-2* in the abstract.

Below are my comments and minor suggestions for you to consider when making some revisions to the text.

Specific comments

(1) Results, lines 88-89: The timing of sex determination in embryos of *Ostrinia furnacalis*, which is given here and in several other places in the results as “during the

period of sex determination”, is referred to Fukui et al. (2023). As this is important for a better understanding of your results, could you define this period? Do you mean the period from the first appearance of the male-specific splice variant of *Ofdsx* (12 h) to the completion of the feminization and masculinization processes (48 h)?

(Authors' reply)

As this Reviewer pointed out, we defined the sex determination period as the period when the sex-specific splicing of *Ofdsx* is first observed after 12 hours post-oviposition (hpo) and is established at 48 hpo. This was added in the revised manuscript (line 97–98).

(2) Fig. S3a: Strains D, E, F and α , β , γ are not mentioned in the Methods and should therefore be explained in the figure legend. What is the difference between the strains (D, E, F) and (α , β , γ)? Yes, fragments of the *Offem* locus were amplified in all female samples, but especially in the (α , β , γ) strains, several bands can also be seen in the male DNA samples. Make this clear.

(Authors' reply)

Strains D, E, F and strains α , β , γ were established from different founder moths and separately maintained. And multiple bands amplified from the male sample of strains α , β , and γ were non-specific. These were included in the figure legend of Fig. S3.

(3) Table S2: The *Ofznf-2* copies identified in chromosome 24 in the genome sequenced by Dai et al. (2024) have the same length as those in genome sequence version 2, where the copies, however, are located on chromosome 28. It is obvious that they are the same copies, but that there is chaos in the chromosome numbering. For *O. furnacalis*, did you use the numbering according to the silkworm where chr 28 matches chr 26 and chr 24 matches chr 28+31 in the *O. furnacalis* assembly of Dai et al. (2024)? See Fig. 2B in Dai et al. (2024). Is it possible that chr 28 in the version 2 genome sequence is identical to chr 24 in the Dai et al. (2024) assembly? The same applies to Fig. S8. Make this clear.

(Authors' reply)

We used the chromosome numbering according to that of *Bombyx mori*, thus chr. 24 in Dai et al. (2024) corresponds to chr. 28 in our newly assembled version #1 and version #2 genomes. This was added in Table S2, S5, and Fig. S8 in the revised manuscript.

(4) References in Supplementary Tables 2, 3, 4: Some of the references given in these

tables are not used in the main text and therefore not listed in the References of the main text. I recommend listing each reference used in these tables under the corresponding supplementary table.

(Authors' reply)

As pointed out by this Reviewer, we added the reference lists under the STables.

(5) Fig. 4a legend (lines 779-780): I think that Wol- should stand for *Wolbachia*-uninfected and Wol+ for *Wolbachia*-infected. Wol+(Tet), which stands for tetracycline-treated *Wolbachia*-infected, should also be explained, as well as the red (and one pink) arrows.

(Authors' reply)

As pointed out by this Reviewer, we added the explanations for Wol-, Wol+, Wol+(Tet), and the red (or pink) arrows in the figure legend of Fig. 4.

(6) Discussion, lines 195-198: Please check references 6, 7 and 8 for the *Wolbachia* sex-determining role in the terrestrial isopod *Armadillidium vulgare* and the butterfly *Eurema mandarina*. The references given here refer to publications on *O. furnacalis* and *O. scapulalis*.

(Authors' reply)

We corrected these points in the revised manuscript (line 204–207).

(7) Discussion, lines 243-245: I believe that this is not the third, but the fourth case of a primary sex determiner that has been molecularly identified in Lepidoptera: *Fem* in *Bombyx mori*, *PxyFem* in *Plutella xylostella*, *Fet-W* in *Lymantria dispar* (see Moronuki et al. 2025), and *OfFem* in *O. furnacalis*.

(Authors' reply)

We corrected this point in the revised manuscript (line 254).

(8) Methods, line 357: Please check the reference to “*Wolbachia*-infected *O. furnacalis* embryos”. You refer to No. 11 (Harumoto and Lemaitre, 2018), but this publication is about *Spiroplasma* in *Drosophila*.

(Authors' reply)

We corrected this point in the revised manuscript (line 371).

(9) Methods, lines 366 and 378: Please check the reference to “LC-MS/MS experiments”. You refer to No. 11 (Harumoto and Lemaitre, 2018), but the LC-MS/MS analysis is not described there. The same applies to this reference to “Western blotting and immunoprecipitation”.

(Authors’ reply)

We corrected these points in the revised manuscript (line 380).

(10) References, lines 712-716: No. 83 is not given in the text, but “DeepConsensus” is mentioned in line 422 and referred to No. 74. Please check it.

(Authors’ reply)

We corrected these points in the revised manuscript. Ref#83 was cited in the legend of Fig. S14.

Minor suggestions

Fig. S2: Although the meaning of “Wol – ” and “Wol+(Tet)” can be deduced, it is better to explain it in the figure legend. The same applies to Fig. S4.

(Authors’ reply)

As pointed out by this Reviewer, we added the explanations for Wol-, Wol+, and Wol+(Tet), in the figure legend of Fig. S2.

Fig. S5: “the” before “*OffFem* piRNA” should be written in normal font.

Fig. S7(b, c) legend: expression

Fig. S8 legend: reference (41)

(Authors’ reply)

We corrected these points in the revised manuscript.

L129: I suggest using “orthologs” instead of “homologs” because there are no data on the function of most znf-2 proteins in the species given in this phylogenetic cladogram.

(Authors' reply)

We agree to this Reviewer's comment that there are no data on the functions of most znf-2 proteins in the species given in this phylogenetic cladogram. Because we just used the data as the sequences with significant homology to znf-2, we think it is better to use "homologs" instead of "orthologs" according to the definition of homolog/ortholog: https://www.nlm.nih.gov/ncbi/workshops/2023-08_BLAST_evolution/ortho_para.html.

Fig. S9, Fig. S13, and Fig. S17 legends: Correct "EGFP-" to "GFP-".

Fig. S15: (a) Mapping of small RNA-seq reads ... (b) Mapping of RNA-seq reads ...

L281: Fig. S6 [in this line, you refer to fig. S5, but this figure does not show positions of the primers]

L282 and L396: Table S3

L384-386: The text starting with "The Methods section ..." should be removed.

L465: Table S1

L496: Biol. Lett.

L570: Correct the publication year 2024 to 2025 in Moronuki et al.

(Authors' reply)

We corrected these points in the revised manuscript.

L774: In Fig. 3c, the green and red dots, highlighting the GFP- and mCherry-fused protein bands, look more like arrowheads or triangles than dots.

(Authors' reply)

As suggested, we changed dots to arrowheads in the revised manuscript.

We thank again Prof. František Marec for his helpful suggestions and encouraging comments.

Reviewer #2:

The manuscript titled “Complete transition from chromosomal to cytoplasmic sex determination during prolonged *Wolbachia* symbiosis” by Fukui et al is reviewed. This study comprehensively investigates the molecular mechanism of sex determination of *Ostrinia furnacalis* with and without infection of *Wolbachia* and provides strong evidence that male-killing *Wolbachia* hijacks the feminizing piRNA function in *Ostrinia furnacalis*. I appreciate the amount of effort in this work and its achievement and only have some minor suggestions to the manuscript.

(Authors' reply)

We thank this Reviewer for his/her encouraging comments.

Line 63: This is optional since it has been mentioned in the discussion but I was wondering if such sex determination in the wild could cause any intraspecific diversification. So I think having information about the infection rate here in the introduction may provide clear picture about how important such sex determination mechanisms are to the evolution of the species.

(Authors' reply)

As suggested by this Reviewer, we included *wFur* prevalence in Japan and modified this part (around line 63 of the original manuscript) in the introduction section (line 63–70).

Line 182: I find this figure (S16) very helpful in summarizing the whole picture, I think it would be better to have this figure in the main paper but not supplementary.

(Authors' reply)

We thank this Reviewer's suggestion. Fig. S16 is a combined picture of Fig. 3d and Fig. 4e, both of them are essential for easier understanding of the results shown in Fig. 3 and Fig. 4, respectively. If Fig. S16 is included in the main text, the context becomes redundant, thus we would like to use this figure as the supplement.

Line 214-216: Here comes inference about genetic drift, I guess it is related to the next section about the prevalence but I think it would be better to have more context about how the fixation of the mutation to the evolutionary history of the population in Japan.

(Authors' reply)

As suggested by this Reviewer, we added a sentence about founder effects to understand

this part easier.

Line 311-318: I was a bit confused here as I thought *OfFem* was identified based on the expression difference, but how the “expression analysis of the *OfFem* locus” section comes after the identification of *OfFem*? I wonder if merging these two sections could make it easier for readers to understand the flow of the work.

(Authors’ reply)

As suggested by this Reviewer, we merged these two sections and added a sentence to understand the flow easier.

Line 333: was there blast criterion applied in blastp? If so, please specify.

(Authors’ reply)

As pointed out by this Reviewer, we added the details of blast search as follows: BLASTp search was performed via the NCBI BLAST homepage (<https://blast.ncbi.nlm.nih.gov/>) with default parameters, using as the query the partial protein sequence of Ofzfn-2 (“MCKDWVRGTCARGAACIYAHLDKDKQLKGVYRFCRDFENDRCERQVCYFV HATTFEKEHFFRTAFLPPHALHHLKT”) (line 344–348).

Line 400: were these samples known for their sex? Are they from the lab breeding lines?

(Authors’ reply)

We used female larvae of laboratory strains. This was added in the revised manuscript (line 411).

Line458: I wonder if it is required to lower “-s” here, I assume the samples are quite inbred and should have low heterozygosity? If now, maybe it will be better to provide the evaluated heterozygosity and to make sense the “-s” parameter change.

(Authors’ reply)

Thank you for this insightful comment. We agree that the rationale for adjusting the assembly parameters warrants further clarification. Although our samples were derived from inbred lines, k-mer based heterozygosity estimates were not extremely low. For example, the assembly version (#1) based on short-read data indicated a genome-wide heterozygosity of approximately 0.74%, and the assembly version (#2) HiFi-read-based

estimate is approximately 0.68%. These values suggest that residual heterozygous regions persist in the genome and may complicate haplotype separation and purging during assembly. In fact, using the default hifiasm parameters (including `-s 0.55`) for version #2, we obtained a primary assembly of 539 Mb in total length, which substantially exceeds the estimated haploid genome size (~470 Mb). While we note that this estimate does not account for heteromorphic sex chromosomes—which are treated as homologous in the estimate and may increase the true genome size slightly—the observed assembly length still appears overly inflated. BUSCO analysis also revealed a large number of duplicated genes (295), suggesting that some haplotypes were assembled as separate contigs, contributing to redundancy. Informed by the hifiasm FAQ, which recommends lowering the `-s` value when the primary assembly is larger than expected or when purging appears insufficient, we explored several `-s` settings. We found that setting `-s 0.30` substantially improved assembly quality by reducing this redundancy and producing a cleaner representation of the genome. This adjustment notably reduced the total assembly size and lowered the number of duplicated BUSCOs from 295 to 22. While this more aggressive purging introduced a few minor artifacts requiring manual curation, we judged the trade-off to be acceptable.

Following your suggestion, we have now clarified this empirical optimization process in the Methods section by adding the following sentence: The "-s" parameter was optimized by comparing assembly statistics from several trial runs, with the primary goal of reducing redundancy caused by unresolved haplotypes in the primary assembly.

Thank you again for your helpful suggestions.

Reviewer #3:

This study reports fascinating research that will be of wide interest with respect to the evolution of insect sex determination systems and of symbiont reproductive manipulations. The authors achieve two objectives; firstly, they identify a new primary Lepidoteran feminising factor in *O. furnacalis* ‘*OfFem*’, a W-linked piRNA that silences a constitutively expressed masculinising factor, and secondly, they demonstrate that the *OfFem* locus has been lost in a *O. furnacalis* lineage carrying the male-killing *Wolbachia* strain *wFur*. These are important, novel contributions, explaining convincingly how ‘cytoplasmic sex determination’ has arisen in this species. The research is of a very high standard, robustly conducted and the results convincing. The data are clearly presented, the manuscript is well written and is for the most part easy to follow, with an excellent discussion.

(Authors’ reply)

We thank this Reviewer for his/her encouraging comments.

It would be useful to orient the reader to provide in the introduction a short overview of how widespread is the *wFur* strain in *O. furnacalis* in Japan, and where it occurs what is the typical prevalence. Are there any *wFur* lineages that have produced 1:1 sex ratios on tet treatment to remove *Wolbachia*?

(Authors’ reply)

To the best of my knowledge, it is unclear how *wFur* had spread in *O. furnacalis* in Japan, and where it had occurred first in Japan. In addition, we do not know Japanese male-killing *Wolbachia* strains in *O. furnacalis* whose depletion restore 1:1 sex ratios.

Specific comments:

Line 108: states that there are 15 repeats of the *Ofzmf-2* target, but Fig. S5 seems to suggest that there are 17 repeats. Please clarify.

(Authors’ reply)

As pointed out by this Reviewer, we corrected this point (17 repeats) in the revised manuscript (line 117).

L214 'it must overcome random genetic drift' - could there be a selective advantage associated with not expressing the W-linked feminizing factor in the presence of *Wolbachia*?

(Authors' reply)

We thank this Reviewer for this comment. We think a loss-of-function mutation in a W-linked feminizing factor is unlikely to provide any selective advantage. This was added in the revised manuscript (line 223).

Fig. 3b

It is difficult to gauge the level of interaction between OfMasc and the Ofzfn-2 variants from these images. Firstly, since the interpretation depends on co-localisation, a 'merge' panel for the fluorescent images could be provided. In addition, the images of cytoplasmic/nuclear co-localisation of Ofzfn-2-E with OfMasc and OfMascNLS in Fig S13 are more convincing of interaction than the images in Fig 3b. and thus Fig 3b could be replaced with Fig S13 in the main text – especially if images can be added showing that Ofzfn-2-L does not show similar nuclear/cytoplasmic co-localisation.

(Authors' reply)

To understand the co-localisation of OfMasc and the Ofzfn-2 variants clearer, we changed Fig. 3b as new data, and the previous Fig. 3b was moved to supplementary data as Fig. S13a in the revised manuscript.

Fig. S3

what the different strains in panel A refer to – there does not seem to be an indication of what they represent anywhere in the ms.

(Authors' reply)

We added the details of strains D, E, F and strains α , β , γ in the figure legend of Fig. S3.

Fig. S4C

Since the presence of *Offem* piRNAs is dependent on the host lineage rather than the *Wolbachia* infection status per se, rather than using Wol-, Wol+(Tet) designations it would be clearer to replace with 'Wol- lineage', 'Wol+ lineage (tet-treated)'. Similarly for the Figure legend of C, emphasise host lineage: e.g. 'in the ovary of tetracycline-treated moths from the *Wolbachia*-infected lineage'.

(Authors' reply)

As pointed out, we corrected the label and legend of Fig. S4C in the revised manuscript.

Minor / Typos

Fig. S7 legend: 'expression' missing e.

Line 255: misspelling of *O. furnacalis*

227 'both protein homologs' would be clearer

232 exhibited a little positive effect – remove little or replace with small

(Authors' reply)

We corrected these points in the revised manuscript.

Thank you again for your helpful suggestions.

Reviewer #4:

(Authors' reply)

We thank this Reviewer for his/her co-reviewing this manuscript.

Point-by-point responses to the reviewer's comments

Reviewer #1:

Well done! In the revised manuscript, the authors have satisfactorily addressed all my comments and suggestions. I look forward to the publication of this excellent study.

(Authors' reply)

We thank Prof. František Marec for reviewing this manuscript and giving encouraging comments.

Note: I have found two minor problems that can be fixed in the further editing of this article. These are:

(1) Fig. 4a legend (line 796): The explanations for Wol-, Wol+ and Wol+(Tet) were added as suggested in my comment #5, but the explanations for Wol- and Wol+ were incorrectly switched [see “Wolbachia-infected (Wol –), uninfected (Wol+)”]. I believe that Wol- should stand for Wolbachia-uninfected and Wol+ for Wolbachia-infected.

(Authors' reply)

As suggested by this Reviewer, we corrected this point in the revised manuscript.

(2) L255-256: At the end of this sentence there could be references to the three previously identified primary sex determinants to clarify it as follows “This marks the fourth instance of a primary sex determiner being molecularly identified in Lepidoptera (18, 19, 30).“

(Authors' reply)

As suggested, we added three references at the end of this sentence in the revised manuscript.

We thank again Prof. František Marec for his helpful suggestions and encouraging comments.

Reviewer #2:

We thank this Reviewer for his/her encouraging comments and helpful suggestions.